# Self-Consciousness as a Construction All the Way Down

**DOI:** 10.3390/bs14030200

**Published:** 2024-03-01

**Authors:** Massimo Marraffa, Cristina Meini

**Affiliations:** 1Department of Philosophy, Communication and Performing Arts, Roma Tre University, 00144 Rome, Italy; massimo.marraffa@uniroma3.it; 2Department of Ecological Transition and Sustainable Development, University of Eastern Piedmont, 13100 Vercelli, Italy

**Keywords:** narrative self, physical self, self-consciousness construction, subjective identity, theory of mind

## Abstract

Contemporary mind and brain sciences provide theories and data that seem to confirm a hypothesis about human nature that we might formulate as follows. Human life is conditioned by a need that is no less important than elementary biological needs (such as survival and reproduction) or universal forms of social competition: the need to build and, indeed, defend a *subjective identity* whose solidity and clarity are the foundation of our intra- and inter-personal equilibrium and therefore of psychological well-being and mental health. In this article, distancing ourselves from a neo-Cartesian position still prevalent in the philosophy of mind and approaching instead the outcomes of contemporary cognitive sciences, we sketch the complex interweaving of the cognitive, emotional, and affective elements that are constitutive of subjective identity, with a focus on the role played in self-identity construction by Theory-of-Mind abilities. We will suggest that, at every stage of self-construction, individuals engage in processes of understanding others that have a largely innate basis. In this perspective, a mature self-awareness is somewhat secondary to the knowledge of others, an evolutionarily refined acquisition primarily serving as a defense mechanism.

## 1. Introduction

Individual identity, when considered in its *objective* aspect (i.e., “for others”), is what makes persons recognizable, or rather, what makes them unmistakable. By this, everyone is categorized by others: male or female, young or old, single or married, of good or precarious social status, native or foreigner, and so on. *Subjectively* (i.e., experienced “from the inside”), on the other hand, identity is knowing who one is and knowing how to define and describe oneself, and to a large extent, it is also knowing how to “tell” oneself, autobiographically.

In this article, we sketch the complex interweaving of the cognitive, emotional, and affective elements that are constitutive of subjective identity, with a focus on the role played in self-identity construction by Theory-of-Mind abilities. In drawing this sketch, we will make very little recourse to the conceptual analysis that characterizes the analytical philosophy of self-consciousness and personal identity; instead, we practice an empirically oriented philosophy of mind (or rather a theoretical psychology) that takes as its model and builds on Peter Carruthers’ work on self-knowledge (on which see below).

## 2. The Self-Identity Construction as a Search for the Self That Tends towards the Synthesis of the Various Layers of the Personality

Since the post-war period, the subject of self-identity has made an unstoppable march in the psychological sciences. In the enormous amount of literature that has been produced, we enumerate three points that we believe could form the basis for developing a neo-Jamesian approach to the self. Firstly, the emergence of the thesis of an inseparable link between self-description of identity and self-consciousness. Secondly, dynamic psychology and developmental psychology have shown that the construction of the affective life, throughout the life cycle, is primarily the construction of an identity that is both well-defined and accepted as valid. Thirdly, social psychology has brought to light our constant negotiation of the validity of our identity in exchanges with others.

The thesis of an inseparable link between self-description of identity and self-awareness is clearly formulated in Dan McAdams’ interpretation of William James’s seminal theory of the self as constituted by the couple <I, Me>.

In 1985, in *Power, Intimacy, and the Life Story*, McAdams [1] proposed a theory of identity development centered on the concept of narrative. A few years later, Katherine Nelson [2] presented a perspective on the ontogeny of narrative capacity that was very close to McAdams’ approach. Subsequently, McAdams’ narrativist theory of identity became a reference paradigm in personality development studies [3].

The foundations of McAdams’ theory lie in Erik Erikson’s studies on identity in adolescence, in the tradition of the Study of Lives inaugurated by the works of Alfred Adler, Henry Murray, and Silvan Tompkins, and, indeed, in the theory of the duplex self formulated by James in *Principles of Psychology* [4].

According to James, McAdams says, the I-self is not a something but “is really more like a verb; it might be called ‘selfing’ or ‘I-ing’, the fundamental process of making a self out of experience” ([5], p. 302). The Me-self is instead “the primary product of the selfing process”; it is “the self that selfing makes” ([5], ibid). The Me exists as an evolving collection of self-attributions (James’ material, social, and psychological selves) that result from the selfing process. It is “the making of the Me that constitutes what the I fundamentally is” ([6], p. 162). Self-consciousness is then viewed as a *process*, by which we mean the self-representing of the “subject” naturalistically viewed as a system consisting of interacting mechanisms operating at different (social, individual, neural, and molecular) levels [7].

Thus interpreted James’ theory clearly establishes the aforementioned inextricable link between self-descriptive identity and self-awareness. James views self-consciousness as the knowledge of being there in a certain way, a self-description, an identity-building. This contrasts with a conception of self-consciousness that is classically formulated by Kant; the consciousness of self can be captured in a *pure* state, independently from the consciousness of existing in a certain way: “I am conscious of myself, not as I appear to myself, nor as I am in myself, but only that I am”, he writes in the *Critique of Pure Reason* (B157). To James, in contrast, self-consciousness cannot be conceived as a self-awareness that is primary, elemental, and simple, preceding any other form of knowing; he thinks that there can be no consciousness of self without knowledge of self.

The point is crucial: one does not know that one is without knowing who one is; we only know that we are there insofar as we know that we are there *in a certain way*, that is, with particular features, as a describable identity; there is no consciousness of existence without there being a description of self and therefore without there being a description of identity ([8], p. 139). We believe that data from developmental psychology are decisive in this respect; observing and studying infants during the second year of age, we realize that there is no difference between the construction of their self-awareness (at first corporeal and physical, but then also psychological and introspective) and the construction of their identity.

Within this framework, the most advanced form of self-identity description is “an internalized and evolving life story” ([9], p. 527); it is “the broad narrative of the Me that the I composes, edits, and continues to work on” ([6], p. 169). The life-story format infuses the tangle of autobiographical memories with “some semblance of unity, purpose, and meaning” ([9], p. 527). That is, individuals give meaning to their life through narrative structures (characters, roles, scenes, scripts, and plots) that make the Me take the form of “an internalized drama” ([6], p. 169). This narration of the self is therefore a process that synthesizes, integrates, and unifies:

With respect to the I, the self functions as a unifying process through which subjective experience is synthesized and appropriated as one’s own. On the side of the Me, the process of appropriating experience as one’s own results in a reflexive conception of self (the me that the I constructs), and such a reflexive product may itself express unity and purpose. Identity in the me is the extent to which the Me can be arranged (by the I) as a unifying and purpose-giving story. For contemporary adults, therefore, the synthesizing I-process creates unity in the Me by fashioning a self-defining product that ideally assumes the form of an integrative life narrative ([10], p. 56).

Overall, by defining the construction of narrative identity as a unifying, integrative, synthesizing process, McAdams is not an isolated voice at all; instead, he places himself in an influential tradition that runs through developmental psychology, dynamic psychology, and personality psychology. Although differing in many respects, concepts such as Werner’s orthogenetic principle, Piaget’s organizational tendencies, Loevinger’s Ego development, and Jung’s individuation, share the idea that human experience tends toward a fundamental sense of unity in that human beings apprehend experience through an integrative selfing process ([10], p. 57).

Furthermore, McAdams [11] locates the unity produced by autobiographical narrative in a personological framework. Narrative identity is then seen as a layer of a personality hinged on two further layers: (i) *dispositional traits* that are “broad internal dimensions of personality thought to account for general consistencies in behavior, thought, and feeling observed across situations” and (ii) *characteristic adaptations* that are “goals, plans, projects, values, possible selves, and other contextualized features of personality capturing individual differences in motivation” ([9], p. 519).

During personality development, internalized and evolving life stories are layered on top of characteristic adaptations, which in turn are layered on top of dispositional traits. This layering process can be *integrative*: the selfing process can succeed in synthesizing traits, abilities, purposes, values, and experiences into a life story with meaning:

Traits capture the actor’s dramaturgical present; goals and values project the agent into the future. An autobiographical author enters the developmental picture, in adolescence and emerging adulthood, to integrate the reconstructed past with the experienced present and envisioned future ([11], p. 226).

The process of selfing then takes the form of the Jungian concept of individuation: a process of integration, i.e., of coherent unification of one’s psychological characteristics.

## 3. The Bodily Self as the Minimal Self

The most basic form of identity self-description is *bodily* self-awareness.

Numerous experimental data show that from the very first weeks of life, self-specifying proprioceptive/kinaesthetic information is precociously available to infants, e.g., [12,13]. These findings have been interpreted as evidence in favor of the thesis that long before the acquisition of a conceptual and objective form of self-consciousness, *a pre-reflective sense of ownership for one’s body* is already present in preverbal infants [14]. To this thesis, we oppose a more cautious reading of these same findings, arguing that postnatal infants are immersed in a subjectivity that they are unable to objectify; they are already agents but do not yet know it, because in place of a unified representation of their body as recognizable as “their own” body, they possess only fragmented and incomplete perceptions of their bodies.

As the mirror self-recognition test attests [15], it is only in the second year of life that infants become able to construct a body image of themselves as *an entire object* while at the same time considering this image as *a subject*, i.e., as an active source of self-representation. This is a new type of object of consciousness: the object is in fact the subject himself. This—we claim—occurs through mediation with the caregiver within the attachment environment. Bodily self-awareness is viewed here as a cognitive acquisition that requires seeing oneself through the eyes of the other, i.e., identifying oneself in someone who is looking at us. As Winnicott puts it, “[i]n individual emotional development *the precursor of the mirror is the mother’s face*” ([16], p. 149). The other is an alter-ego, the mirror function through which the body recognizes itself and is accepted as a whole. Note that this hypothesis assumes that the process of constructing a physical self is rooted in already established basic capacities for understanding others, such as those made possible by the precursors of Baron-Cohen’s mindreading system ([17]; see also the more elaborate version in [18]).

Self-awareness in its basic form is therefore the representation of a physical self. In other words, we access the idea of existing only because we become able to identify ourselves “in flesh and blood”, namely, insofar as we “know” that we are individuals with certain characteristics, which are primarily physical, physiognomic, and bodily characteristics.

## 4. The Phenomenology of Basic Emotions Takes Shape in Bodily Reflexivity

A more advanced phenomenon than bodily self-awareness is introspective reflexivity. The modern definition of this concept is Locke’s definition: persons as rationally and morally conscious and responsible individuals are such because they are capable of introspectively appropriating their actions, i.e., representing and recognizing them as their own, thereby considering their meaning. But to do this, agents must be able to represent not only bodily actions, but also the intentions and affections that they produce within themselves and therefore must be able to represent their inner world, objectifying the latter but at the same time making it their own. This is introspection: knowing that one is considering, objectively, the various aspects of one’s subjectivity. The place of reflexivity is here no longer only the body as a real dimension, but the mind as an *internal* virtual dimension.

As in the case of bodily self-awareness, we argue that the mind as an internal virtual dimension is the result of a constructive process that rests on the domain-specific understanding of others’ psychological functioning. This is completely at odds with the “strong intersubjectivist view” which assumes a pre-wired introspective subjectivity ([19], p. 29). This is a Cartesian view insofar as it sees knowledge of one’s own mental states as somehow *primary*, “with knowledge of the mental states of other people emerging later (in both phylogeny and ontogeny), dependent on one’s awareness of one’s own mental life” ([20], p. 9). The most elaborate version of such a perspective is the simulationist position defended at length by Alvin Goldman [21], but which we believe has been refuted by Peter Carruthers’ theory of introspective self-knowledge (see Section 5 below).

In accordance with this anti-Cartesian intent, we believe that a cogent argument can be made in favor of a socio-constructivist view of the ontogeny of the earliest form of *psychological* self-awareness—an *affective* bodily self-consciousness. In this perspective, infants endowed with a precocious (and arguably innate) capacity to recognize basic emotions in others’ expressions [22] must develop an introspective awareness of the discrete basic emotion states which are initially unconscious automatisms. In fact, there are no data to suggest that in the early stages of development, infants’ affective life is introspectively visible. It is the well-tuned relationship with significant others that is the factor that promotes the path that, starting from a *core affect* phenomenology (an initially undifferentiated structure of valence and arousal dimensions), leads to the internalization of a repertoire of discrete emotions. This inaugurates a new dimension of subjectivity, which will gradually expand until it shapes children’s entire mental lives.

One of the most robust hypotheses about the construction of emotional introspection, the social biofeedback model of parental affect mirroring [23,24,25], offers elements and insights to reconstruct a development path composed of the following main steps:(1)The first form of differentiation between infants’ object field and their subjective world takes place by *valence* dimensions: positive valence gives rise to feelings of acceptance–pleasure–security–incorporation, while negative valence gives rise to feelings of rejection–insufficiency–anguish–expulsion [26]. At this initial point, no phenomenology is associated with the experience of basic emotions.(2)The distinctive phenomenological component of basic emotions is added in a microsocial context, largely through interpersonal affect mirroring. Infants, who possess innate and precocious competence in recognizing the basic emotional expressions of others, notice that adults typically address marked expressions to them. Through social biofeedback and the process of contingency detection that is part of it, infants gradually come to anchor marked parental expressions to their own as yet undefined emotional experience, thus producing second-order representations of their affective states.(3)The complete internalization of discrete emotions in infants’ subjective world is produced when the phenomenology of basic emotions takes shape in bodily reflexivity: bodily self-images become *affective* bodily self-images. In other words, early first-person emotional mentalization consists of the acquired capacity to group primary somatic data (the visceral and proprioceptive cues that are triggered when the child is in an affective state and expresses it) into categories of discrete emotions, which are then attributed to oneself and thereby internalized in one’s subjective world [27].

## 5. Expanding Introspective Space

Self-consciousness is initially the phenomenological seat of emotional life but must then expand considerably to become what James calls the “spiritual self” and Locke assumes as a precondition of personhood.

During this further development, the social dimension retains its crucial role. The expansion of the virtual space of the mind occurs mainly because children, under the socio-communicative pressure of caregivers and peers, become capable of turning onto themselves a repertoire of socio-cognitive skills that were originally focused on others. These abilities consist primarily of those Theory-of-Mind skills we use spontaneously in everyday life. It can then be said that with this development, “the proper domain of the human mindreading becomes ontogenetically extended to include in its actual domain the mind of one’s own self as well” ([28], p. 74).

In Theory-of-Mind literature, different interpretations of the data emerging from the research on understanding the nature of beliefs have been proposed. We think Victoria Southgate’s [29] “altercentric bias” is a very plausible hypothesis; children’s spontaneous attention to the perspective of others may already have the nature of a true theorization of other minds. This supports the view that, in the epistemic domain as well as in the domain of discrete emotions, the first space of mentalistic analysis that opens up for humans is directed towards other people. In fact, in the face of a sense of self that must be built over time, a very large body of literature attests to innate human capacities for understanding the mental processes of others, starting with the false belief test that children pass, in the implicit version, around the age of one year [30].

On these grounds, we make the hypothesis that the socio-cognitive skills originally focused on others, and then involved in self-construction, are mainly based on two neurocomputational systems: the first is the one underlying mindreading, and the second is the system underlying sociomoral thinking.

The mindreading system is the key construct in Peter Carruthers’ theory of introspection [31,32], which belongs to a philosophical and psychological tradition that accounts for self-knowledge in terms of *parity between the first- and third-person* [33]; in contrast to “inner sense” theories of self-knowledge ([32]), the parity tradition holds that the processes by which we acquire knowledge of our minds are the same processes by which we acquire knowledge of other people’s minds.

In Carruthers’ view, the parity thesis is restricted to a subset of mental states, namely, propositional attitude events (henceforth “thoughts”). Although we can have non-interpretive access to our perceptual and imagistic states, self-attribution of thoughts is always a process of self-interpretation subserved by a faculty of mindreading that exploits the same sensory channels that we utilize when working out other people’s mental states.

Carruthers [32] makes a strong case for his theory of introspective self-knowledge by taking a position on the studies on confabulation in cognitive neuropsychology and social psychology. One of the central predictions of Carruthers’ interpretive sensory access (ISA) theory of self-knowledge is that “cases of confabulation should occur” ([32], p. 365). Inner sense theories [21,34] explain confabulation by postulating two methods, one introspective and one interpretative: it is true that, in certain circumstances, individuals interpret themselves based on a theory (which may give rise to confabulatory discourses); but it is also true that, on other occasions, they enjoy access to their mind that is direct and non-interpretive. Yet, it is problematic for inner sense theories to exactly define the circumstances in which one has direct access to one’s propositional attitudes and those in which recourse is made instead of self-directed mentalization. Conversely, assuming that knowledge of one’s thoughts is based on an interpretative process fed by sensory and behavioral data and driven by a naïve psychology theory, ISA theory enables one to predict confabulation effects whenever such data are misleading or the theories used for interpretation are inadequate.

In its initial version, ISA theory gave third-person mindreading functional and phylogenetic priority over introspection, but Carruthers ([31], p. 167) felt there was insufficient data to pronounce on the possibility of a developmental priority. Carruthers [20], however, believes that there is a case to be made that “awareness of the mental states of other people emerges first in ontogeny […]. Self-knowledge, on the other hand, results from turning one’s mind-reading abilities on oneself” ([20], p. 9).

Insofar as Carruthers’ theory gives third-person mentalization an ontogenetic priority, it is a natural complement to the hypothesis on the development of emotional understanding that was the topic of Section 4. However, Carruthers’ theory does not go into the details of the process of turning the mindreading system on the self; moreover, the focus of his analysis is only on thoughts. Accordingly, his theory must be integrated in at least two ways.

In the first place, since Carruthers’ theory views inner speech as providing the essential data for thought self-attribution, it can be profitably compared with Lev Vygotsky’s [35] theory of inner speech development.

Believing that private speech is a product of the internalization of communicative language, Vygotsky regarded it as a fundamental tool for the self-regulation of behavior, thus aligning perfectly with current research on working memory and self-reflective functions ([36]; see also Morin’s [37] analysis of introspective reports made by Jill Bolte Taylor, the neuroanatomist who, struck by a left hemispheric stroke, lost her inner speech). And although, in light of contemporary experimental data, one can exclude a constitutive role of language (in its dual public and private dimensions) in the development of naïve psychology, one cannot doubt its role as a factor of enrichment and—we believe—progressive internalization.

Secondly, the narrow focus on epistemic mindreading may have led Carruthers to underestimate the strong influence that the development of sociomoral thinking exerts on the process of turning our mindreading skills upon ourselves, hence the need to also examine the literature that investigates the connection between the construction of introspection and sociomoral knowledge.

In search of the reason as to why we have the self-deceptive intuition that there is introspection for our thoughts, Carruthers [31] deems plausible Gazzaniga’s [38] and Wilson’s [39] hypothesis that a Cartesian belief in the self-transparency of minds “may make it easier for subjects to engage in various kinds of adaptive self-deception, helping them build and maintain a positive self-image” ([31], p. 138).

This openness of Carruthers to the psychodynamic theme of defenses is necessary insofar as the ISA theory draws heavily on the confabulation data from the literature on cognitive dissonance and causal attribution, and such data can hardly be separated from the topic of the construction and maintenance of “a positive self-image”. In social psychology, the defense of self-image (closely linked to the defensive use of causal attribution), and the rationalizing function of cognitive dissonance (as well as social attitudes in general, and stereotypes and prejudices in particular), are conceived as constitutive elements of an interpersonal and social reality which is rich in structures of self-deception, namely, defensive constructions resulting from mental processes in which cognitive aspects cannot be well separated from the affective ones.

Clarification is important. The focus of ISA theory is *not* on introspective self-knowledge construed as “awareness of oneself as an ongoing bearer of mental states and dispositions, who has both a past and a future” ([40], p. 14). Its focus is on knowledge of one’s current mental states, and this knowledge “is arguably more fundamental than knowledge of oneself as a self with an ongoing mental life” (ibid.). In fact, after arguing that the emergence of introspection is a by-product of the evolution of mindreading, Carruthers notes that this fact does not preclude that introspection might “have come under secondary selection thereafter, perhaps by virtue of helping to build and maintain a positive self-image, as Wilson […] suggests” ([31], p. 128). Thus, first introspection emerges as a competence to self-attribute one’s current thoughts; but it is only when it has become “knowledge of oneself as a self with an ongoing mental life” that Wilson’s hypothesis of the self-defensive nature of introspection can be proposed to explain the secondary adaptation process to which it is subjected. It is only then in fact that the turning of our mindreading faculty upon ourselves can be seen as the construction of an introspective self-consciousness which is primarily the construction—in the socio-communicative interaction with caregivers and other social partners—of a subjective identity that is both well-defined and accepted as valid.

As we have seen, the infant/caregiver interaction is made first of preverbal exchanges; it then becomes an interaction made of words, descriptions, designations, and evaluations of the person. During verbal conversations, the *evaluative descriptions* are progressively internalized by children, becoming part of their subjective identity. At the heart of this internalization process is a primary need for *interpersonal validation* of one’s subjective identity. Children cannot ascribe concreteness and solidity to their self-awareness if it does not possess at its center, and as its essence, a description of identity that must be clear and, indissolubly, “good”. One’s mental equilibrium is based on one’s feeling of solidly existing as a self that is *worthy of being loved*, which consequently constitutes for the subject what more than anything else must be defended.

This need for interpersonal validation of subjective identity leads us to consider the close link between the construction of interiority and ethics. Indeed, alongside the now decades-long investigation into children’s naïve psychological competence, more recent investigations into the development of sociomoral reasoning have flourished [41,42,43,44], revealing an early and rapidly increasing tendency to judge individuals in positive or negative terms, and to evaluate actions as, from time to time, obligatory, permitted, forbidden. In a synergistic relationship, naïve psychology and sociomoral competence are likely to provide children with essential tools to prepare them for progressively more expert and effective social navigation [45].

## 6. Conclusions

In this article, we draw on the psychodynamic, socio-cognitive, and developmental literature to outline a perspective on the development of subjective identity, aiming to unfold the potentialities of the Jamesian theory of the duplex self. A fully socio-constructivist analysis of self-development emerged, which stands as antithetical to the hypothesis of a pre-reflective self-consciousness which is typical of the neo-phenomenological perspective, e.g., [46,47]. The self (James’ pair <I, Me>) has been conceptualized as a psychobiological unifying process (the process of ‘self-ing’ or ‘I-ing’) incessantly building and updating self-representations, from bodily to narrative self-representation. Fundamental point: the first form of introspective self-consciousness is made possible by the construction of a representation of the self that is both bodily and emotional.

Our proposal has a psychodynamic character. The construction of subjective identity is an ongoing and inexhaustible search for a socially validated self-description, and the intertwining of cognitive and affective dimensions characterizing such a process results in an identity that is not given once and for all; it is rather something perpetually rebuilt and actively reconfirmed, something perennially precarious. Such precariousness makes the theme of self-identity construction inseparable from that of self-identity defense [48,49].

This way of conceiving self-awareness in its various forms aims to contribute to an anti-Cartesian agenda, which rejects the claim that one’s awareness of one’s own mental life has a functional, phylogenetic, and ontogenetic priority over the knowledge of the mental states of other people [20]. From its earliest stages, cognitive processes aimed at understanding others, and likely an expression of the functioning of innate domain-specific neurocomputational systems, are involved in the construction of the self. The Cartesianism implicit in many accounts of our knowledge of mental states is thus overturned in favor of a thesis that sees awareness of the mental states of other people emerging first in ontogeny, and self-knowledge resulting from turning one’s hetero-directed competencies (among which are Theory-of-Mind abilities) on oneself.

## Data Availability

The original contributions presented in the study are included in the article, further inquiries can be directed to the corresponding authors.

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
