# Peer review of "Self-Consciousness as a Construction All the Way Down"

_behavsci, 2024, doi:10.3390/bs14030200_

Round 1

Reviewer 1 Report

Comments and Suggestions for Authors

Review Report

This article presents a constructivist analysis of self-development, challenging the notion of pre-reflective self-consciousness prevalent in the neo-phenomenological perspective. Instead, self-consciousness is seen as an ongoing construction process, starting with automatic processing of object representations and progressing through body awareness, self-awareness, introspection, and narrative identity. This perspective aims to oppose Cartesianism by emphasizing that cognitive processes involved in understanding others contribute to the construction of the self from its earliest stages. It suggests that each stage of individual development is primarily focused on comprehending others, overturning the primacy of self-knowledge in mental states.

Overall, I commend the authors for presenting a theory that is both comprehensive and intricate. As someone who is not well-versed in the psychological or empirically oriented aspects of philosophy of mind, I must admit that I find it challenging to grasp some of the terminology used. In order to enhance the accessibility of the essay, I would suggest that the authors define key technical terms, thereby making the content more self-contained. In my training within the analytic philosophy tradition, we strive to ensure that our essays are understandable even to outsiders and non-academic readers. While I am uncertain if the same expectation holds true in the field of behavioral science, where the authors are presumably working, it is generally beneficial to provide at least concise explanations for crucial terms.

In order to improve clarity, it would be highly beneficial for the authors to more clearly distinguish their own viewpoints from those of others. Throughout the essay, it becomes challenging to discern which points are being put forth by the authors themselves and which are attributed to other scholars. For instance, Section 2 (pages 1 to 3) appears to primarily consist of a presentation of various theories proposed by scholars, lacking in-depth analysis. This confusion may arise from a blending of theory delineation with the authors' own comments on these theories. To address this, it would be valuable for the authors to explicitly state their stance on these theories, providing explanations for why they have chosen to include specific quotations and engaging in critical analysis, critique, or comparative discussions. In the field of analytic philosophy, active sentence constructions are commonly utilized, such as "Scholar Y proposes this. I agree, as the reason behind my agreement is Z. On the other hand, Scholar A puts forth a proposition that I find problematic, given the reasons of N. Now, we propose bla bla bla, motivated by the reason of U." Such an approach enables a clearer distinction between the authors' own contributions and those of other scholars, fostering a more engaging and analytically rigorous essay.

It leads me to wonder if the author has addressed the concept of primitive self-consciousness put forth by philosophers like Sydney Shoemaker, John Perry, Lynne Rudder Baker, and others. According to this perspective, one can possess self-awareness without having explicit knowledge of their own identity. For instance, a patient who has lost all memories may still engage in reflective thoughts such as "Where am I?" or "What will I eat for lunch?", despite lacking knowledge of personal details such as their name, occupation, family members, friends, and other aspects of their identity.

The author writes, “McAdams says, the I-self is not a something but “is really more like a verb; it might be called ‘selfing’ or ‘I-ing’, the fundamental process of making a self out of experience” ([5], p. 302). The Me-self is instead “the primary product of the selfing process”; it is “the self that selfing makes” ([5], ibid).” It appears that the authors endorse this perspective. However, I am curious to know if the authors consider the presence of a subject of experience, a distinct entity that possesses experiences, within this framework. For many philosophers, positing such a subject of experience is necessary to account for the ownership of experiences, enabling the integration and coherence of experiences into a unified whole, before delving into the process of constructing bodily, narrative, or psychological identities from these experiences. Is there a subject that undergoes the initial creation of self-identification within this framework?

I am uncertain whether the authors really oppose the Cartesian assertion regarding the primacy of self-knowledge concerning one's mental states, as they explicitly mentioned in the conclusion of their work. Your theory appears to be a developmental account of self-knowledge, focusing on how individuals acquire different levels of self-awareness. In contrast, the Cartesian claim pertains to the metaphysical nature of our mental states, specifically that when experiencing pain, we possess immediate knowledge of this fact. Interestingly, your theory could potentially support the Cartesian claim by demonstrating how self-knowledge is cultivated through interactions between infants and others.

Moreover, it is worth noting that the Cartesian self is conceived as a substance, a thinking entity that possesses experiences. However, your discussion primarily centers around the construction of one's identity, encompassing the processes of self-understanding and self-identification within society and the broader world, particularly through narrative means. To me, it appears that Descartes and the authors may be addressing different aspects at varying levels, and it is not entirely clear why one perspective is incompatible with the other. Providing further elaboration would be beneficial in clarifying this point. Alternatively, the authors might consider removing the claim that their approach aims to contribute to an anti-Cartesian agenda, unless they can explicate the specific incompatibilities more explicitly.

Given the authors' target audience in the field of behavioral science, readers may be intrigued to know if there are empirical methods available to test the hypothesis presented. For example, the authors propose that the process of developing a physical self is grounded in preexisting foundational abilities for comprehending others. Can this proposition be subjected to empirical testing? If the authors have any specific ideas or potential approaches in mind, it would be valuable to mention them, even if they are still at a preliminary stage. Of course, the authors are welcome to honestly acknowledge that these ideas are in their early stages of development.

Your paper has prompted me to contemplate solitary animals: those that live alone. I am curious to know the authors' perspective on how these animals develop a sense of embodiment, establish self-identity, self-consciousness, self-images, and so on. Is your theory exclusively applicable to humans? If social interactions may not be crucial for the development of self-consciousness in non-human animals, what accounts for the divergence in the development of self-consciousness between non-human animals and humans?

Author Response

We thank both reviewers for the attention they devoted to our paper.

Regarding Reviewer 1’s suggestions for improvement, we have made interventions in the paper and highlighted our insertions in the text.

More specifically, addressing each of the points raised:

  • We have introduced a brief definition of technical terms and crucial passages that were previously implicit, such as the relevance of the mirror task. As correctly noted, this was important to enhance the accessibility of the essay for those not well-versed in empirically-informed philosophy of mind.
  • In various passages, starting from the Abstract and the Introduction, we have more accurately distinguished our viewpoint from those of other philosophers and cognitive scientists. Concerning Section 2 (mentioned by Reviewer 1), it is dedicated to analyzing various theoretical viewpoints. Throughout the paper, we have explicitly attributed the various positions to their respective authors and clarified where they converge or diverge from the view we develop in subsequent sections.
  • We did not specifically address the concept of primitive self-consciousness put forth by philosophers like Shoemaker, Perry, and Rudder Baker. Instead, we focused on authors such as Gallagher and Zahavi, who explicitly reference the same scientific literature we engage with. Additionally, we introduced a passage in the reviewed version that explicitly delineates William James’s view of self-consciousness as self-description from Kant’s idea of it as a pure state.
  • More generally, our theoretical stance has been elucidated in a new passage in the Introduction: “[…] we will make very little recourse to the conceptual analysis that characterizes analytical philosophy of self-consciousness and personal identity; rather, our article can be viewed as an exercise in “theoretical psychology,” following the lines of Peter Carruthers’ theory of self-knowledge.”
  • Yes, we endorse McAdams’ idea of the Me as the primary product of the selfing process, and more generally, the Jamesian image of the self as a process. Consistent with this view, we believe that there is no subject of experience independent of the process of constructing bodily, narrative, or psychological identities.
  • As pointed out by Reviewer 1, “Descartes and the authors may be addressing different aspects at varying levels, and it is not entirely clear why one perspective is incompatible with the other.” Nevertheless, we do not believe we should remove the claim that our approach aims to contribute to an anti-Cartesian agenda, as there is an extensive debate in empirically-informed philosophy of mind, and more broadly in cognitive sciences, framed in these terms. Therefore, our analysis should be understood within this theoretical context: our ambition is to contribute to this ongoing debate.
  • Admittedly, empirical research on self-consciousness is a highly complex endeavor, perhaps the most challenging in the psychological domain. The reasons are manifold, starting from the inherent complexity of the processes themselves, not to mention the convergence between the subject and object of analysis. Despite these unavoidable complexities, we believe that our paper is empirically informed: many authors we mention have conducted significant empirical work on the topic, proposing relevant empirical models and tasks (e.g., Gergely and Watson, Baron-Cohen, Morin, Southgate, Russell, Nichols).
  • Nineteenth-century ethological and systemic research has shown how the relational dimension is crucial for animals in general, to the extent that we should question whether genuinely solitary animals exist. In any case, what we argue is that human development, and arguably that of other animals possessing certain neural properties (such as mammals and animals with a limbic system), occurs in a social environment that plays a primary role in the developmental process itself. This does not amount to denying that a purportedly solitary animal could, in principle, exploit other resources to construct at least bodily self-consciousness. That said, we would emphasize that 1) there is a distinction between having an objectual consciousness of our body and recognizing our body as our own, as occurs in two-year-old human beings. Possibly, then, other animals have the first but not the second form of bodily self-consciousness and 2) it is not evident at all that solitary animals possess any form (either objectual or subjective) of more sophisticated self-consciousness, such as psychological self-consciousness.

Reviewer 2 Report

Comments and Suggestions for Authors

This paper makes a case against neo-phenomenalism by contending that self-consciousness is never pre-reflective, but instead emerges in dialogue with others through the internalisation of social interactions that are inherently reflective. I must say from the outset that I am not convinced of the arguments presented, as I feel there is a cursory dismissal of pre-natal and early-natal abilities in lines 126-131. The assertion is not supported by references and, although the constitutive role of language is addressed later, it does not consider the obvious confound that expressing one's subjectivity in language is not the same thing as experiencing one's subjectivity pre-verbally. I think what is presented is a weak argument, but not an invalid one. The ideas are presented cogently and consistently, so I see no barrier to publication. 

Author Response

Thank you very much for taking the time to review our manuscript.